

# A Kullback-Liebler divergence-based representation algorithm for malware detection

Faitouri A. Aboaoja[1,2], Anazida Zainal[1], Fuad A. Ghaleb[1], Norah Saleh Alghamdi[3], Faisal Saeed[4] and Husayn Alhuwayji[5]

[1] Faculty of Computing, Universiti Teknologi Malaysia, Johor Baru, Johor, Malaysia
[2] Faculty of Education-Elgobbah, University of Derna, Libya, Elgobbah, Barka, Libya
[3] Department of Computer Sciences, College of Computer and Information Sciences, Princess Nourah Bint Abdulrahman University, Riyadh, Saudi Arabia
[4] DAAI Research Group, Department of Computing and Data Science, School of Computing and Digital Technology, Birmingham City University, Birmingham, UK
[5] Higher Institute of Science and Technology, Qarabulli, Higher Institute of Science and Technology, Qarabulli, Tripoli, Libya

Corresponding authors
Faitouri A. Aboaoja,
faitouri@graduate.utm.my
Norah Saleh Alghamdi,
nosalghamdi@pnu.edu.sa

## ABSTRACT

**Background**. Malware, malicious software, is the major security concern of the digital realm. Conventional cyber-security solutions are challenged by sophisticated malicious behaviors. Currently, an overlap between malicious and legitimate behaviors causes more difficulties in characterizing those behaviors as malicious or legitimate activities. For instance, evasive malware often mimics legitimate behaviors, and evasion techniques are utilized by legitimate and malicious software.

**Problem**. Most of the existing solutions use the traditional term of frequency-inverse document frequency (TF-IDF) technique or its concept to represent malware behaviors. However, the traditional TF-IDF and the developed techniques represent the features, especially the shared ones, inaccurately because those techniques calculate a weight for each feature without considering its distribution in each class; instead, the generated weight is generated based on the distribution of the feature among all the documents. Such presumption can reduce the meaning of those features, and when those features are used to classify malware, they lead to a high false alarms.

**Method**. This study proposes a Kullback-Liebler Divergence-based Term Frequency-Probability Class Distribution (KLD-based TF-PCD) algorithm to represent the extracted features based on the differences between the probability distributions of the terms in malware and benign classes. Unlike the existing solution, the proposed algorithm increases the weights of the important features by using the Kullback-Liebler Divergence tool to measure the differences between their probability distributions in malware and benign classes.

**Results**. The experimental results show that the proposed KLD-based TF-PCD algorithm achieved an accuracy of 0.972, the false positive rate of 0.037, and the F-measure of 0.978. Such results were significant compared to the related work studies. Thus, the proposed KLD-based TF-PCD algorithm contributes to improving the security of cyberspace.

**Conclusion**. New meaningful characteristics have been added by the proposed algorithm to promote the learned knowledge of the classifiers, and thus increase their ability to classify malicious behaviors accurately.

# INTRODUCTION

Malicious software, also referred to as malware, is a program or piece of code that is intended to access a system without the user's permission and perform harmful actions (*Sharma & Sahay, 2014*). The majority of cyber vulnerabilities and attacks, such as global threats, advanced persistent threats (APTs), sensitive data theft, remote code execution, and distributed denial of service (DDoS) attacks, are driven by malware (*Dixit & Silakari, 2021*). The frequency, level of complexity, and financial harm caused by malware infections have been rapidly growing in recent years (*Aslan, Samet & Tanrıöver, 2020*; *Gunduz, 2022*). A Symantec Internet Security Company report (*Symantec, 2015*) revealed that each day in 2014, around one million new malware were launched into the Internet. In 185 different countries, Kaspersky reported 277,646,376 malicious Internet attacks during the third quarter of 2017 (*Hashemi, Samie & Hamzeh, 2022*). In their study, (*Catak et al., 2021*) claimed that between 2011 and 2020, the quantity of new malicious software significantly increased. Recently, a report from (*AV-TEST n.d*) demonstrated that the number of malicious software has increased significantly, as explained in Fig. 1 by which the increased ratio of new malware and potentially unwanted applications (PUA) is shown during the period of time between 2008 and 2022. This is because some available malware creation toolkits allow even the non-expert to adopt several forward strategies like polymorphism and metamorphism using the help of obfuscation techniques to generate sophisticated malware (*Kakisim, Nar & Sogukpinar, 2020*).

To deal with this massive amount of malware, *Ali et al. (2020)*, *Kim, Shin & Han (2020)*, *Finder, Sheetrit & Nissim (2022)* and *Nunes et al. (2022)* move forward using a sandbox-based analysis approach, by which thousands of malware are analyzed daily instead of analyzing the malware statically. Several studies (*Shijo & Salim, 2015*; *Darshan & Jaidhar, 2019*; *Yoo et al., 2021*) stated that dynamic analysis offers more reliable detection capabilities than static analysis. On the other hand, malware authors use immediate evasion techniques as a defense against dynamic analysis (*Kim et al., 2022*). By analyzing 45,375 malware samples, *Galloro et al. (2022)* concluded that the use of evasion mechanisms has increased among malware by 12% over the past ten years, and 88% of malicious software can perform new evasion behaviors rather than the older ones. Evasive malware instances either imitate legitimate behaviors or violently interrupt the execution in sandboxed execution conditions (*Bulazel & Yener, 2017*; *Alaeiyan, Parsa & Conti, 2019*; *Or-Meir et al., 2019*; *Afianian, Niksefat & Sadeghiyan, 2019*; *Mills & Legg, 2020*). Additionally, *Galloro et al. (2022)* in their work, reported that evasion behaviors have picked up in both malware and benign instances because those evasion techniques have been originally developed for a legitimate purpose, such as to prevent reversing, and protect intellectual property.

One of the most intelligent strategies for malware to evade suspicion is to mimic legitimate behaviors. The developed malware detection models now have a significant

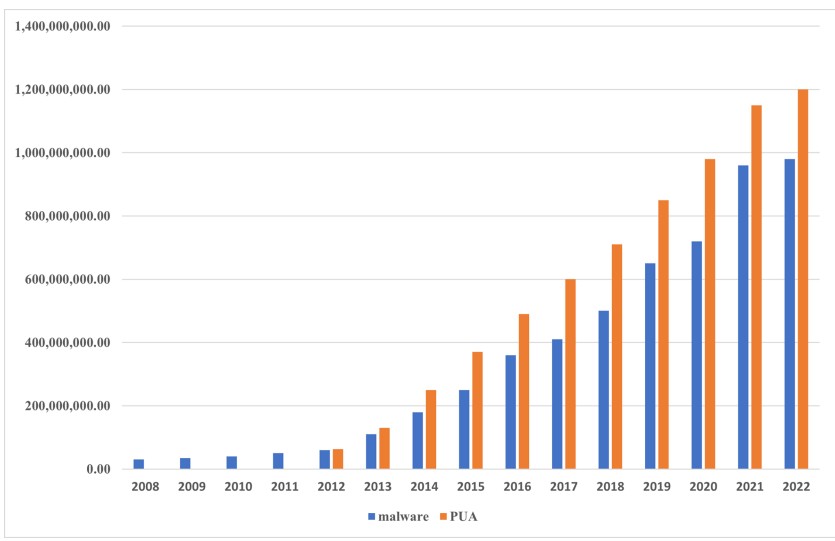

**Figure 1 Malware statistical.** The number of new malware and potentially unwanted applications (PUA) created between 2008 and 2022.

challenge as a result of this mimicking behavior (*Amer, El-Sappagh & Hu, 2020*). The overlapping in malware and benign behaviors by which each class can be characterized causes a serious challenge for the developed models, which may suffer from misclassification trouble, and thus the overall detection accuracy is negatively affected by increasing the false positive and negative rates (*Yang & Lim, 2021*; *Sun et al., 2021*). This challenge comes from the fact that these behaviors have to be constructed in the form of a dataset in which the features that characterize each class are extracted and represented. Therefore, an inaccurate features representation technique can be the root cause of the model confusion situation, which leads to the misclassification problem.

Therefore, malware detection researchers have attempted to develop malware classification models by which malicious and benign behaviors are accurately represented using several feature representation techniques. A well-known TF-IDF technique is imported from the information retrieval field and used for representation purposes by several malware detection researchers (*Zhang et al., 2019*; *Ali et al., 2020*; *Li et al., 2020a*; *Li et al., 2020b*) to represent the extracted features in the form of weight-based vectors. Furthermore, several studies (*Wang & Zhang, 2013*; *Xue et al., 2019*; *Xiao et al., 2020*; *Al-Rimy et al., 2020*; *Qin, Zhang & Chen, 2021*) have been carried out to develop various feature representation techniques by enhancing the concept of the traditional TF-IDF technique and boost its capability to accurately represent the extracted feature. However, the primary principle of these techniques has been built based on the main concept of the traditional TF-IDF technique, by which the probability distributions of the features in each class are not considered when the IDF is calculated. Otherwise, the appearance of each feature in the documents is considered regardless of the classes to which these documents belong. Therefore, such representation does not necessarily accurate represent the sharing features that belong to the mimic legitimate behaviors performed by malware,

or the evasion techniques offered by benign and low weights can be assigned for those kinds of features. As a result, an enhanced feature representation technique is required to improve the detection accuracy of the proposed malware classification model and decrease the false positive and negative rates.

To this end, this paper proposes a Kullback-Liebler Divergence-based Term Frequency—Probability Class Distribution KLD-based TF-PCD algorithm that accurately represents the extracted features, including the shared features in the form of weight-based vectors. Our proposed algorithm is developed based on the fact that the probability distribution of using evasion behaviors differs between malware and benign classes (*Maffia et al., 2021*). In contrast to existing solutions which rely on the appearance of the features in the documents, the KLD-based TF-PCD algorithm calculates the weights according to the TF of each feature in the concerned document multiplied by the difference between the probability distributions of that feature in malware and benign classes. The difference between the probability distributions is calculated using the Kullback-Liebler Divergence method. The intuition is that even the sharing features can be useful when their probability distributions are not similar in malware and benign classes, thus as long as the dissimilarity in the probability distributions of the feature is bigger, the weight of that feature is greater. In summary, this paper makes the following contributions:

1. A Kullback-Liebler Divergence-based Term Frequency-Probability class Distribution (KLD-based TF-PCD) algorithm was proposed to represent the extracted features using numerical weights which were generated based on the probability distributions of the features in malware and benign classes.

2. Strengthen the feature space by enhancing the usefulness of the features that have appeared differently in malware and benign classes by using their distribution in each class.

3. A comprehensive experimental evaluation was carried out to demonstrate the improvement that the KLD-based TF-PCD algorithm had made.

4. Conducting an in-depth comparative analysis between the utilized KLD-based TF-PCD algorithm and the recent related feature representation techniques in terms of the obtained classification accuracy and other detection metrics.

The paper is organized as follows: In the "Related Work" section, this paper presents the recent related work. Then, a full description of Kullback-Liebler Divergence tool and its usage in malware analysis and detection field are introduced in the "Kullback-Liebler Divergence" section. In the "The Proposed Method" section, the methodology that was followed to design and develop the proposed algorithm is illustrated. The experimental design, including the dataset, performance measures, obtained results, comparison, and significance test are shown in the "Experimental Design" section. After that, the experimental results are analyzed and discussed in the "Analysis and Discussion" section. The "Conclusion" section concludes the paper and presents future research directions.

## RELATED WORK

Several studies (*Burnap et al., 2018*; *Naz & Singh, 2019*; *Kakisim, Nar & Sogukpinar, 2020*; *Ahmed et al., 2020*; *Catak et al., 2020*; *Galloro et al., 2022*; *Nunes et al., 2022*; *Finder, Sheetrit*

*& Nissim, 2022*) have been done for malware detection based on the signature, behavioral, and heuristic approaches. Static features such as opcodes and PE header data have been extracted by *Naz & Singh (2019)* and *Kakisim, Nar & Sogukpinar (2020)* to identify the characteristics of malware, while (*Burnap et al., 2018*; *Ahmed et al., 2020*; *Galloro et al., 2022*; *Nunes et al., 2022*; *Finder, Sheetrit & Nissim, 2022*) have observed and recorded the malware behaviors during the run-time like API calls, system calls, and machine activity metrics. For the aim of malware detection and classification, most studies trained machine learning techniques using the extracted features with several feature representation techniques by which the extracted features are represented in the form of numerical vectors to be understandable by machine learning algorithms (*Aboaoja et al., 2022*).

To enhance the performance of malware detection and classification models, various studies were conducted to suggest solutions by which the weaknesses of the feature representation techniques could be mitigated. Those solutions can be categorized into three main kinds, binary-based, frequency-based, and weight-based. According to the presence or absence of each extracted feature in each document, the values 1 or 0 are assigned to that feature in the generated vector. *Shijo & Salim (2015)* and *Sihwail et al. (2019)* created binary vectors by which each extracted n-gram feature was represented using 0 or 1 to characterize each sample in their datasets using its corresponding binary vector. A support vector machine (SVM) classifier was trained and evaluated using the generated binary vectors. Both studies achieved satisfactory detection accuracy of 0.985 and 0.987, respectively. *Banin, Shalaginov & Franke (2016)* trained (K-NN) k-nearest neighbor and (ANN) artificial neural network classifiers using binary-based vectors which were constructed to represent the most frequently dynamic n-gram features. The developed models provided an accuracy of 0.989. However, the legitimate behaviors, which are frequently injected into the developed malware by malware writers to circumvent the analysis attempts, may be represented in malware binary vectors as malicious characteristics by which the false positive rate can be increased (*Singh & Singh, 2018*; *Aboaoja et al., 2022*).

In frequency-based features, on the other hand, the similarity between malicious behaviors can be measured based on the frequency (occurrence times) of each extracted feature to identify multiple variants of the same family. The frequency-based representation technique generates vectors in which the frequency count of each feature in the document is assigned to represent the performed behaviors. Furthermore, the frequency-based representation techniques were developed based on the assumption that there is a difference between the frequency of the performed function in malware and benign classes (*Yewale & Singh, 2017*). As VBasic-based malware was a concern of *Ali et al. (2020)* in their study, the VBscript samples were explored to identify certain functions, methods, and keywords together with their frequencies to build frequency-baes vectors. Using n-gram technique, the authors of *Galal, Mahdy & Atiea (2016)* extracted API calls, and then frequency-based vectors of each n-gram were generated to describe the file properties. However, the popularity of employing obfuscation techniques to produce irrelevant features using dead code insertion, instruction reordering, and equivalent code replacement can be the most common source of generating unrepresentative frequency vectors (*Mirzazadeh, Moattar & Jahan, 2015*; *Elsersy, Feizollah & Anuar, 2022*).

Moreover, to mitigate the shortcomings of the above-mentioned feature representation techniques, several studies such as *Belaoued et al. (2019)* and *Ali et al. (2020)* selected the most important features based on the weights that were calculated using the traditional TF-IDF technique, while (*Li et al., 2020a*) used TF-IDF technique to select and represent the proposed feature set. Other studies (*Xue et al., 2019*; *Xiao et al., 2020*; *Al-Rimy et al., 2020*; *Qin, Zhang & Chen, 2021*) developed the traditional TF-IDF to propose enhanced TF-IDF techniques by which the obtained features were represented using more accurate weights. *Xue et al. (2019)* proposed a malware classification model that connected a convolutional neural network (CNN) trained on static features and the random forest (RF) trained on dynamic features *via* a probability scoring threshold. Furthermore, the API-based n-gram features by which the RF classifier was trained were selected and represented using the weights that were calculated utilizing the developed document frequency-inverse document frequency DF-IDF technique.

To solve the problem of the computational complexity of graph-based malware detection models, *Xiao et al. (2020)* developed an API-based graph malware detection model. The constructed graph is divided into fragment behaviors and the crucial behaviors have been extracted and represented by joining TF-IDF and IG information gain methods. The extracted crucial behaviors are used to train machine learning classifiers. *Al-Rimy et al. (2020)* proposed feature extraction scheme for accurately extracting representative ransomware attack features from the pre-encryption phase. To represent the extracted features, an annotated term frequency-inverse document frequency (aTF-IDF) technique has been developed. The proposed (aTF-IDF) penalized the general-purpose API calls that come after the pre-encryption phase since the developed technique is capable of observing the API calls before and after the pre-encryption phase. Therefore, the problem of insufficient characteristics through which the traditional TF-IDF increases its weights when it computes the IDF is mitigated.

A few studies have employed the Inverse class frequency ICF component during the representation stage by few studies. While (*Wang & Zhang, 2013*) used inverse class frequency ICF rather than inverse document frequency IDF to highlight features that appear in fewer classes than those that appear in more classes, *Qin, Zhang & Chen (2021)* combined IDF and ICF to introduce the term frequency-(inverse document frequency and inverse class frequency) TF-(IDF & ICF) technique in which API sequences were represented as weight-based vectors. Consequently, machine learning classifiers were trained using those generated vectors to provide an accuracy of 0.979 by the LR logistic regression classifier.

However, the traditional TF-IDF technique suffers from inaccurately weighted features. This is because the weights are calculated by multiplying TF (the frequency rate of the feature in one document) by IDF (describe how rarely the feature appears in all the documents). Therefore, the probability distributions of the features in each class are not considered. Ignoring the probability distribution of the features in each class and settling only on identifying how many the documents in which the features are appeared lead to a decrease in the mean of the features that are appeared in both classes regardless of the frequency of their appearances in each class. This is because the probability distributions

of those features are varied between malware and benign classes (*Maffia et al., 2021*). Moreover, the developed TF (IDF & ICF) is computed through the summation of IDF and ICF components. The ICF component is calculated by dividing the total number of the classes by the total number of the classes that contain the concerned feature. This technique is developed for the multiclassification task. Therefore, in the case of a binary classification task, the ICF component will be abstractly assigned as 1 for all the features that appear in both classes since there is no consideration for the degree to which the frequencies of those features differ in each class. Additionally, the IDF component is calculated without considering the feature distribution in each class. The (DF-IDF) technique seems to be more appropriate for multiclassification models than binary classification models. This is because applying the DF-IDF technique with a binary classification model leads to assigning inaccurate weights for the extracted features since the difference between the distributions of each feature in malware and benign classes has never been identified. The (TF-IDF & ICF) technique is developed based on the concepts of traditional TF-IDF and ICF techniques, and thus the proposed technique suffers from the shortcomings of IDF and ICF components. On the other hand, while the aTF-IDF seems adequate to represent the pre-encryption features of a ransomware attack, its nature makes it inapplicable to represent the evasion behaviors since there are no pre-stage-based features.

To fill the gaps in the current techniques, this paper proposes a feature representation algorithm by which the extracted features can be accurately represented using numerical weights that are calculated with a consideration of the TF of the feature in one document and the degree of difference between the probability distribution of that feature in malware and benign classes. The proposed technique can mitigate the confusion level of the developed model since the weights of the features that appear in both classes will be different as a result of the diversity of the probability distribution of those features in each class. Table 1 shows a summary of the most utilized feature representation techniques in terms of components (how to calculate each part in the developed techniques), and the weakness of each developed technique.

## KULLBACK-LIEBLER DIVERGENCE

A tool called Kullback-Liebler Divergence (KLD) is employed to compare two probability distributions and to measure the difference between them. Information theory and probability theory are where the concept originally developed. The KL divergence represents a non-symmetric measurement to estimate the difference between the probability distributions $p(x)$ and $q(x)$. The KL divergence of $q(x)$ from $p(x)$, denoted $DKL(p(x), q(x))$, is a measure of the information lost when $q(x)$ is used to approximate $p(x)$ (*Sartea et al., 2020*; *University of Illinois, 2021*). $DKL(p(x), q(x))$ is defined in Eq. (1):

$$\text{KLD}\big(p(x), q(x)\big) = \sum_{x \to X} p * \log\left(\frac{p(x)}{q(x)}\right). \tag{1}$$

KLD tool is utilized as a distribution difference measurement tool in several malware detection and classification studies for many purposes, such as distinguishing between

passive and reactive traces (*Sartea et al., 2020*), accelerating dynamic analysis (*Lin, Pao & Liao, 2018*), identifying image changes (*Fargana & Baku, 2019*), measuring the effectiveness of the extracted features (*Zhang et al., 2018*).

Based on bayesian games between a sandbox agent and a malware execution trace, *Sartea et al. (2020)* created an active malware analysis AMA technique. The sandbox achieved an action by which the malware is triggered to perform malicious behaviors, and thus the proposed technique linked each malware family to the notion of the bayesian game types that reflect the malware family triggers. Kullback-Liebler Divergence KLD tool is employed to compute the difference between the distribution of APIs (P) for passive execution traces by which the malware exhibited malicious behavior without any triggers and the distribution of APIs (Q) for reactive execution traces, which represent the malware that needs triggers to present malicious activities. Practically, the mean value of KLD between the distributions of passive and reactive execution traces is computed in the training phase. Therefore, during the test phase, the mean KLD value is used to classify the tested passive and reactive traces.

With a focus on accelerating sandbox-based dynamic analysis, *Lin, Pao & Liao (2018)* proposed an information measurement-based virtual time control (VTC) mechanism in which the clock source is generated based on a pre-identified speed rate. The main aim of the developed VTC mechanism is to record as many system calls as possible within limited resources. Random variables based on Shannon entropy are used to figure out how often system calls occur (system call frequency) in each time interval. Additionally, the KLD tool is used to measure the difference between the distributions of system calls in the current time interval and the next time interval. As long as the KLD value is close to zero due to the low difference ratio between the compared system call distributions owing to the fact that there are no additional new system calls being produced, the speed rate has to be increased to accelerate the execution analysis and thus record new system calls.

Moreover, because new malware has been created by making petty updates to the previously created ones, *Fargana & Baku (2019)* developed a malware detection model based on discovering the changes by which the new malware images can be changed compared to the training images. RGB images are divided into grid blocks with the same dimensions. Furthermore, two Gauss distributions are measured for each grid. The KLD tool is used to estimate the difference between the Guass distributions of two image grids. The Gaussian Mixture Clustering model is utilized to cluster the KLD data to classify the changes that have been made to the images.

In their work, *Zhang et al. (2018)* used the KLD as an evaluation tool to investigate the effectiveness of the extracted sensitive sys calls. The most frequent sys calls in unpacked malware and not in benign code are extracted. Furthermore, the important features are selected using the IG method. The selected features have been used to train and evaluate the proposed principal component initialized multi-layers neural network. To ensure the effectiveness of the extracted sensitive sys calls, the KLD tool is used to measure the distributions of the sensitive sys calls between unpacked malware/benign and packed malware/benign.

**Table 1 Summary of the developed TF-IDFs as feature representation techniques, their components, and weaknesses.**

| Ref | Technique | Components Component1 | Component2 | Weakness |
|---|---|---|---|---|
| *Ali et al. (2020)*, *Li et al. (2020a)*, *Li et al. (2020b)* and *Soni, Kishore & Mohapatra (2022)* | *TF-IDF* | $TF = \frac{\text{number of occurrences of feature (F) in document D}}{\text{number of all features in document D}}$ | $IDF = \log\left(\frac{\text{total number of all documents N}}{\text{number of documents contain feature (F)}}\right)$ | The weights of IDF component are calculated based on the distribution of the features in all the documents without discovering the feature distribution in each class. Thus, the importance of all the sharing features is decreased even if there is differences in their distributions between classes. |
| *Al-Rimy et al. (2020)* | *aTF-IDF* | $TF = \frac{\text{number of occurrences of feature (F) in document D}}{\text{number of all features in document D}}$ | $IDF = \log\left(\frac{\text{total number of all documents } N}{\text{number of documents contain feature (F)}}\right)$ | Although aTF-IDF sounds promising to represent the pre-encryption characteristics of ransomware attack, its architecture prevents it from being used to represent evasion behaviours because there are no pre-stage-based features. |
| | | Normalized $tf = \frac{TF}{\text{length of trace file}}$ | $aTF - IDF = atf \lvert\text{feature}\rvert = \begin{cases} 1, & \text{if pre} \\ 0, & \text{other} \end{cases} * IDF$ | |

*(continued on next page)*

Aboaoja et al. (2023), *PeerJ Comput. Sci.*, DOI 10.7717/peerj-cs.1492

**Table 1** (*continued*)

| Ref | Technique | Components | | Weakness |
|-----|-----------|-----------|-----------|----------|
| | | **Component1** | **Component2** | |
| *Qin, Zhang & Chen (2021)* | *TF-(IDF & ICF)* | $TF = \dfrac{\text{number of occurrences of feature (F) in document D}}{\text{number of all features in document D}}$ | $ICF = \dfrac{\text{total number of classes C}}{\text{number of classes contain feature}}$ | Both IDF and ICF component drawbacks can be surfaced when using the TF-(IDF & ICF) technique on a binary classification task. Regarding ICF, each feature that occurs in malware and benign classes will have a low ICF value regardless of how many it is distributed in each class. On the other hand, the IDF value is computed without considering the distribution of the concerned feature in each class. |
| | | | $IDF = \dfrac{\text{total number of documents D}}{\text{number of documents contain feature}}$ | |
| | | | TF-(IDF & ICF) $= \log_{10}(1 + ICF + \text{In}(IDF))$ | |
| *Xiao et al. (2020)* | *TF-IDF&IG* | *TF-IDF* | *IG* (information gain) | Even though IG values are calculated to measure the probability distribution of each feature in malware and benign classes as described in *Lin, Pao & Liao (2018)*. However, the values that are generated using TF-IDF suffer from the weakness of traditional TF-IDF. Therefore, an overall weight which is computed through multiplying TF-IDF value by IG value inherits the original failure of traditional TF-IDF. |
| | | | TF-IDF & IG $= \alpha$ TF-IDF $+ (1-\alpha)$IG | |

**Table 1** (*continued*)

| Ref | Technique | Components Component1 | Component2 | Weakness |
|-----|-----------|----------------------|-----------|----------|
| *Xue et al. (2019)* | DF-IDF | $DF = \dfrac{\text{number of benign contain feature (F)}}{\text{number of all benign sample} D(k)}$ | $IDF = \log\left(\dfrac{\text{number of all malware samples D}(k)}{\text{number of malware contain feature (F)}} + 1\right)$ | Using this technique with a binary classification model leads to highlight the features which are frequently appeared in both classes as important features. This is because each feature is weighted based on multiplying its document frequency DF in benign class by its inverse document frequency in malware class. Therefore, the obtained weights may not accurate since the difference between the distributions in malware and benign classes has never identified. |

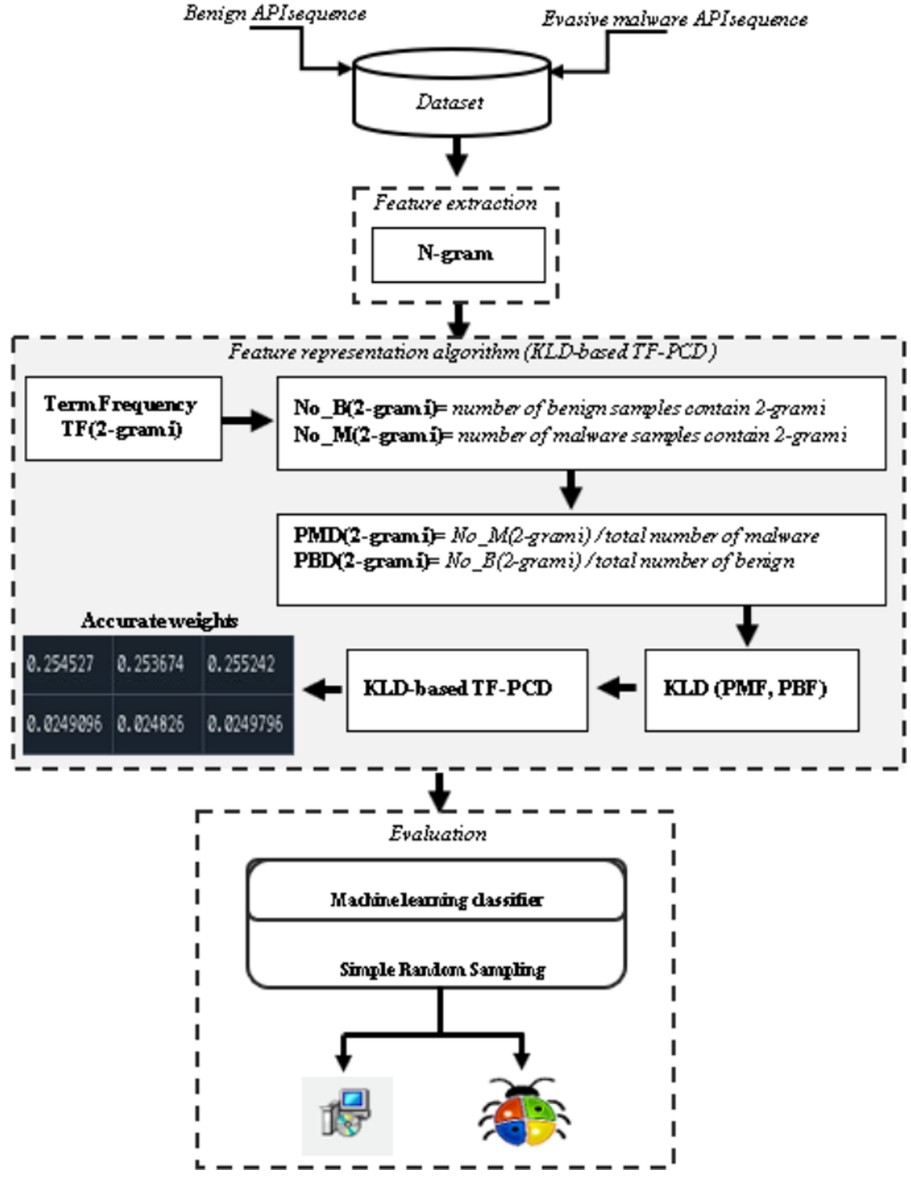

**Figure 2    An overview of the proposed method.**

## THE PROPOSED METHOD

Figure 2 shows an overview of the proposed method. Feature extraction, feature representation, and evaluation are the main three stages that are carried out to evaluate the performance of the proposed KLD-based TF-PCD algorithm. While the feature extraction stage extracts API-based 2-gram features, the developed KLD-based TF-PCD algorithm is utilized in the feature representation stage. Finally, several machine learning techniques are trained and evaluated using features that are represented using the proposed algorithm. The following subsections describe those stages in detail.

## Feature extraction

Several studies (*Fuyong & Tiezhu, 2017*; *Zhang et al., 2019*; *Li et al., 2020b*; *Yang & Liu, 2020*) have suggested the N-gram technique for obtaining N-length or sub-text features from the initial text. From the API sequences of every evasive malicious software sample and benign sample in the database, we extract 2-gram features. Because malware must make several API calls rather than just individual API calls in order to perform malicious activities (*Pektaş & Acarman, 2017*), we chose the API sequence-based N-gram features. As a result, the extracted API sequence-based N-gram features can be beneficial in understanding and modelling malicious behaviour.

## Feature representation

In this subsection, the Kullback-Liebler Divergence-based Term Frequency-Probability Class Distribution (KLD-based TF-PCD) algorithm is proposed to represent the extracted features, including the features that occur in both benign and malware classes, using accurate weight vectors. The drawback of employing the traditional TF-IDF technique is that the distribution of each extracted feature in benign and malware classes is neglected when computing the IDF term. Such neglect leads to calculation of inaccurate IDF values for the sharing features, and thus the overall generated weights are negatively affected by IDF values.

Moreover, mimicking legitimate behavior is a widely used strategy by todays malware (*Bulazel & Yener, 2017*; *Alaeiyan, Parsa & Conti, 2019*; *Or-Meir et al., 2019*; *Afianian, Niksefat & Sadeghiyan, 2019*; *Mills & Legg, 2020*; *Amer, El-Sappagh & Hu, 2020*). In addition, evasion behaviors have been observed in both benign and malicious classes (*Galloro et al., 2022*). Applying the traditional TF-IDF technique to represent the features related to the above-mentioned behaviors will introduce small IDF values, which cause low TF-IDF weights. Therefore, these features can be ignored in the case of using a feature selection technique or using them with close weights, in which the classification model is confused and provides high false positive and negative rates in case feature selection technique is not utilized. This study supposes that these kinds of features can be meaningful features if their probability distributions in both benign and malware classes are exploited since benign and malware classes exhibited evasion behaviors in different probability distributions (*Maffia et al., 2021*).

The KLD-based TF-PCD algorithm tackles this issue by calculating the probability distributions of each feature in both benign and malware classes and then identifying the difference between the probability distribution of every feature in the benign class and the probability distribution of every feature in the malware class using the Kullback-Liebler Divergence tool. The general formula applied to calculate KLD-based TF-PCD weights is shown in Eq. (2).

$$\mathrm{W}\left(n\_\mathrm{gram}_i^j\right) = \mathrm{TF}\left(n\_\mathrm{gram}_i^j\right) * \log\left(-1 * \mathrm{KLD}\left(\mathrm{PMD}\left(n\_\mathrm{gram}_i^j\right), \mathrm{PBD}\left(n\_\mathrm{gram}_i^j\right)\right)\right) \quad (2)$$

where $w\left(n-\mathrm{gram}_i^j\right)$ represents the calculated weight of the $i$th $n$-gram in malware instance $j$, $\mathrm{TF}\left(n-\mathrm{gram}_i^j\right)$ represents the term frequency, or how many times the $n-\mathrm{gram}_i$ is invoked by instance $j$. TF is calculated using the formula illustrated in Eq. (3). The

probability distribution of $n-\mathrm{gram}_i^j$ in malware class and the probability distribution of $n-\mathrm{gram}_i^j$ in benign class are denoted by $\mathrm{PMD}(n-\mathrm{gram}_i^j)$ and $\mathrm{PBD}(n-\mathrm{gram}_i^j)$, respectively. PMD and PBD are computed using the formulas described in Eqs. (4) and 5, respectively. KLD refers to the Kullback-Liebler Divergence tool that measures the difference between the probability distribution of $n-\mathrm{gram}_i^j$ in malware and benign classes. KLD is calculated using the formula shown in Eq. (6).

$$\mathrm{TF}\,(n\_\mathrm{gram}_i^j) = \frac{\mathrm{count}(n\_\mathrm{gram}_i^j)}{\sum_k \mathrm{count}(n\_\mathrm{gram}_k^j)} \tag{3}$$

where $\mathrm{TF}\,(n-\mathrm{gram}_i^j)$ is the frequency of $n-\mathrm{gram}_i$ in malware instance $j$, count $(n-\mathrm{gram}_i^j)$ refers to the number of times $n-\mathrm{gram}_i$ occurs in malware instances $j$, and $\sum_k \mathrm{count}(n-\mathrm{gram}_k^j)$ stands for the total number of all $n$-grams in malware instance $j$.

$$\mathrm{PMD}\,(n-\mathrm{gram}_i^j) = \frac{\mathrm{count\_M}(i,j)}{N(m)} \tag{4}$$

$$\mathrm{PBD}\,(n-\mathrm{gram}_i^j) = \frac{\mathrm{count\_B}(i,j)}{N(b)} \tag{5}$$

where $\mathrm{PMD}(n-\mathrm{gram}_i^j)$ and $\mathrm{PBD}(n-\mathrm{gram}_i^j)$ refer to the probability distribution of $n-\mathrm{gram}_i^j$ in malware class and benign class, respectively. $\mathrm{count\_M}(i,j)$, $\mathrm{count\_B}(i,j)$ represent the number of malware samples that $n-\mathrm{gram}_i^j$ is appeared in and the number of benign samples that contain $n-\mathrm{gram}_i^j$, respectively, while $N(m)$ stands for the total number of malware samples, $N(b)$ denotes the total number of benign samples.

$$\mathrm{KLD}\,(\mathrm{PMD}\,(n-\mathrm{gram}_i^j), \mathrm{PBD}\,(n-\mathrm{gram}_i^j)) = \mathrm{PMD}\,(n-\mathrm{gram}_i^j)$$
$$*\log 2\left(\frac{\mathrm{PMD}\,(n-\mathrm{gram}_i^j)}{\mathrm{PBD}(n-\mathrm{gram}_i^j)} + 1\right) \tag{6}$$

where $\mathrm{KLD}(\mathrm{PMD}(n-\mathrm{gram}_i^j), \mathrm{PBD}(n-\mathrm{gram}_i^j))$ refers to the difference in the probability distributions of $n-\mathrm{gram}_i^j$ in malware class and benign class, $\mathrm{PMD}(n-\mathrm{gram}_i^j)$ means the probability distribution of $n-\mathrm{gram}_i^j$ in malware class, $\mathrm{PBD}(n-\mathrm{gram}_i^j)$ is the probability distribution of $n-\mathrm{gram}_i^j$ in the benign class, the constant 1 is added to prevent the difference value to be zero value.

Unlike traditional and developed TF-IDF utilized by existing malware classification solutions, the KLD-based TF-PCF technique used the differences between the probability distribution of the concerned feature in malware class and the probability distribution of that feature in benign class instead of IDF values.

IDF can be more useful in the natural language processing field (NLP) when a text classification task is required. Our assumption is that the text classification task in the NLP field differs from the text classification task in the malware classification field. Since the texts in the NLP field belong to human languages, the words that frequently appear in all the documents, such as the, for, of, and others, are not meaningful words because they frequently occur to give the same meanings for all the documents, while the texts in the malware classification field were not originally human languages. Otherwise, these texts

belong to malware and benign behaviors. Therefore, the generalization of the concept of IDF on behavioral-based text classification without considering how differently those behaviors have appeared in malware and benign classes can reduce the discrimination of those behaviors.

## Evaluation

This study utilized a variety of machine learning techniques that are extensively employed in the literature, including k-nearest neighbor (KNN), regression trees (CART), Naive Bayes (NB), support vector machine (SVM), artificial neural network (ANN), random forest (RF), logistic regression (LR), and eXtreme Gradient Boosting (XGBoost). Furthermore, 40% of the constructed dataset is randomly reserved for testing purposes as unseen data in order to evaluate these machine learning classifiers, and the remaining 60% has been used to train the chosen machine learning models using the 10-fold-cross validation method. To ensure that all classifiers are evaluated using the exact same data when we compare our suggested model with the relevant work, we used the simple random sampling method to split the dataset into train and unseen test data. Figure 3 shows the training and test stages.

## EXPERIMENTAL DESIGN

In this section, the methodology of validating the developed model is covered. We describe the dataset deployed, identify the specification of the experimental environment, the performance measures used to evaluate the proposed model, and the experimental results of the proposed model have been presented including the comparison with the related studies. Finally, a $t$-test has been done to measure the proposed technique's significance.

### Experiment environment setup

The experiments have been carried out under the environment with specifications as follows: Windows 10 Pro 21H2 is installed on a PC equipped with an Intel(R) Core(TM) i7-4790 CPU running at 3.60 GHz and 16.0 GB of memory size. Python (3.9) libraries using Spyder editor version 5 have been utilized to implement feature extraction, feature representation, and classification techniques that have been developed in this study.

### Dataset

We dynamically obtained API call sequences that represent behaviors of 7208 evasive malware collected from *Galloro et al. (2022)* and *Kirat & Vigna (2015)* as well as 3848 benign samples collected from *Wei et al. (2021)* and the freshly installed Windows 7 operating system to evaluate our suggested algorithm. We established our dataset with this number of samples based on the acceptance number we gained from the literature review in the community of malware detection, which mostly ranged between 1000 and 5000 samples (*Nunes et al., 2019*; *Sihwail et al., 2019*; *Zhang et al., 2019*; *Kakisim, Nar & Sogukpinar, 2020*; *Yoo et al., 2021*; *Nunes et al., 2022*; *Finder, Sheetrit & Nissim, 2022*). The majority of authors either did not make their used datasets accessible or provided broken URL connections (*Amer, El-Sappagh & Hu, 2020*). Consequently, establishing datasets for evasive malware and benign software was certainly not an easy operation. Our experiments

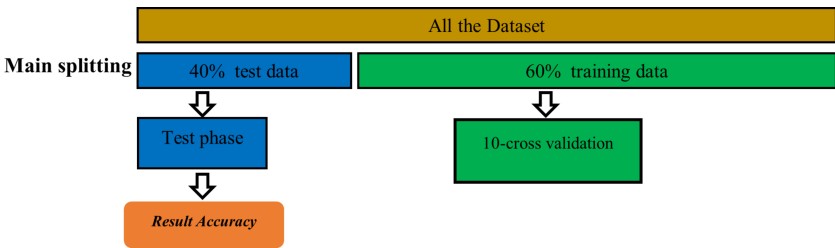

**Figure 3** Training and testing stage.

**Table 2** Evasive malware types and benign files.

| Class | Evasion techniques | Number | Source |
|---|---|---|---|
| Malware | Hardware id-based evasion | 82 | *Kirat & Vigna (2015)* |
| | Bios-based evasion | 6 | *Kirat & Vigna (2015)* |
| | Processor feature-based evasion | 134 | *Kirat & Vigna (2015)* |
| | Exception-based evasion | 197 | *Kirat & Vigna (2015)* |
| | Timing-based evasion | 689 | *Kirat & Vigna (2015)* |
| | Wait for keyboard | 3 | *Kirat & Vigna (2015)* |
| | Not available | 6,096 | *Galloro et al. (2022)* |
| Malware Total Number | | 7208 | |
| | Benign | 2000 | Win7 |
| | Benign | 1848 | *Wei et al. (2021)* |
| Benign Total Number | | 3848 | |

in this work are carried out using the API call sequences dataset that was generated using malware and benign samples mentioned above which have been used in our earlier work (*Aboaoja et al., 2023*). The contents of the used dataset are displayed in Table 2.

## Performance measures

Several performance validation measures, such as the False Positive Rate (FP), False Negative Rate (FN), Accuracy (ACC), Detection Rate (DR), Precision (P), and F-measures (F1), have been applied to assess the effectiveness of the proposed algorithm. Further, FN depicts the proportion of malware classified as legitimate, whereas FP shows the rate of benign classified as malware (*Darshan & Jaidhar, 2020*; *Arslan, 2021*). While the detection rate (DR)/Recall calculates the percentage of malicious samples that are accurately classified using Eq. (7), the accuracy (ACC) assesses the fraction of samples that are effectively recognized using Eq. (8) (*Rostamy et al., 2015*). Moreover, Precision (P) uses Eq. (9) to calculate the ratio of malware samples that are estimated to be a malware across all malicious samples and benign samples that are estimated to be malware. Using Eq. (10), F-measure (F1) sums the mean values of Precision and Detection rate (*Rostamy et al., 2015*).

$$DR = \frac{TP}{TP + FN} \tag{7}$$

$$ACC = \frac{TP + TN}{Tp + TN + FP + FN} \tag{8}$$

$$P = \frac{TP}{TP + FP} \tag{9}$$

$$F1 = \frac{2*Precision*Recall}{Precision + Recall}. \tag{10}$$

## Experimental results

To validate the performance of the proposed Kullback-Liebler Divergence-based Term Frequency-Probability Class Distribution (KLD-based TF-PCD) algorithm, it has been applied on a dataset in which the API call sequence-based 2-gram features were stored to represent the features in the form of a weights-based vector for each instance in the dataset. Furthermore, the dataset has been divided into 60% as training data and 40% as unseen test data using a simple random sampling method. 60% of the dataset instead of 80% or 90% is selected to be the percentage of the training data to guarantee the robustness of the developed model because each improvement in the obtained accuracy means more effectiveness of the proposed model against unseen data points, which have further potential to examine the proposed model as long as the test data size is large. Additionally, the training data was utilized to train multiple machine learning techniques, K-Nearest Neighbor (KNN), Regression Trees (CART), Naive Bayes (NB), Support Vector Machine (SVM), Artificial Neural Networks (ANN), Random Forest (RF), Logistic Regression (LR), and eXtreme Gradient Boosting (XGBoost). To evaluate how each classifier was learned from the weight vectors that were generated using the proposed KLD-based (TF-PCD) algorithm, the test data was utilized to measure the classification performance of each classifier.

Even though the imbalanced dataset is considered one of the problem domains in the malware detection community, it is not our focus in this work. Additionally, the related work compared with the proposed algorithm was trained using imbalanced datasets. However, to estimate the negative effect of an imbalanced dataset, we applied the SMOTE technique to generate balanced training data. The results, using unseen test data, show that the model that trained based on the balanced dataset decreases the accuracy obtained by only 0.004%. The interpretation of the decrement in the accuracy may be referred to as the SMOTE has generated more noises and overlapped features leading to a decrease the accuracy. However, an in-depth investigation is required to study the impact of an imbalanced dataset on classification performance. Such investigation has been left for future work.

Table 3 displays the performance results of the developed classifiers which were trained based on weight vectors generated using the proposed KLD-based(TF-PCD) algorithm. The classifiers' accuracy rates varied from 0.794 for NB to 0.972 for XGBoost. Likewise, the

**Table 3  Experimental results.** Experimental results of the KLD-based (TF-PCD) on dataset with different classifiers.

| Classifier | Acc | FPR | FNR | DR | P | F1 |
|---|---|---|---|---|---|---|
| KNN | 0.898 | 0.130 | 0.086 | 0.914 | 0.930 | 0.922 |
| CART | 0.944 | 0.059 | 0.054 | 0.946 | 0.968 | 0.957 |
| NB | 0.794 | 0.088 | 0.269 | 0.731 | 0.940 | 0.823 |
| SVM | 0.943 | 0.084 | 0.042 | 0.958 | 0.955 | 0.957 |
| ANN | 0.959 | 0.043 | 0.039 | 0.961 | 0.977 | 0.969 |
| RF | 0.956 | 0.066 | 0.032 | 0.968 | 0.965 | 0.967 |
| LR | 0.929 | 0.102 | 0.055 | 0.945 | 0.946 | 0.945 |
| XGBoost | 0.972 | 0.037 | 0.023 | 0.977 | 0.980 | 0.978 |

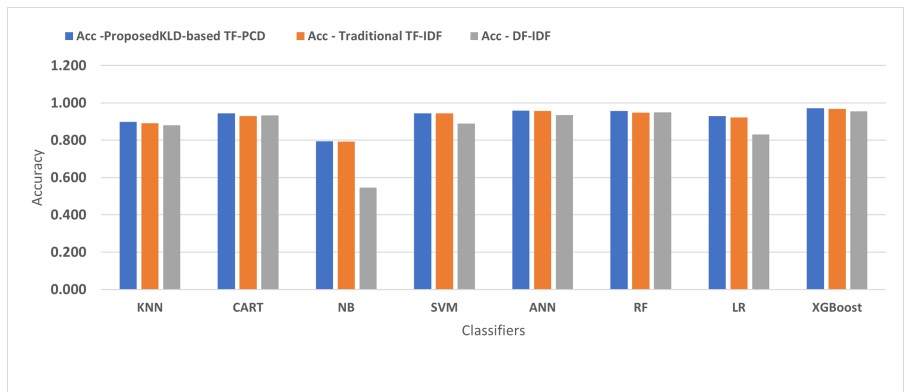

**Figure 4  The comparison of the Detection Accuracy between the proposed KLD(TF-PDF)-based model and the related work-based models.**

F1 rates ranged from 0.823 for NB to 0.978 for XGBoost. The results illustrate that FPR and FNR values dropped together to be 0.037 and 0.023, respectively, with XGBoost. Further, the highest FPR is performed by KNN with 0.130, while NB achieved the highest FNR of 0.269. Regarding the DR, XGBoost introduced the highest value with 0.977, whereas the lowest DR of 0.914 was provided by KNN. For P, the classifiers achieved P rates that ranged between 0.930 for KNN and 0.978 for XGBoost.

Figures 4, 5, 6, 7, 8 and 9 present the comparison of the proposed KLD-based TF-PCD with the traditional TF-IDF, which was widely used in the state-of-the-art (*Belaoued et al., 2019*; *Ali et al., 2020*; *Li et al., 2020a*) and the DF-IDF that was developed by *Xue et al. (2019)*. In terms of accuracy and F-measure, the proposed KLD-based TF-PCD was higher than other techniques for all the classifiers. Furthermore, most of the classifiers with the proposed KLD-based TF-PCD achieved FPR, FNR, DR, and P higher than the traditional TF-IDF and DF-IDF techniques. The best performance compared to all the classifiers was achieved by the XGBoost classifier, which was trained using the weight vectors generated using the proposed KLD-based TF-PCD algorithm.

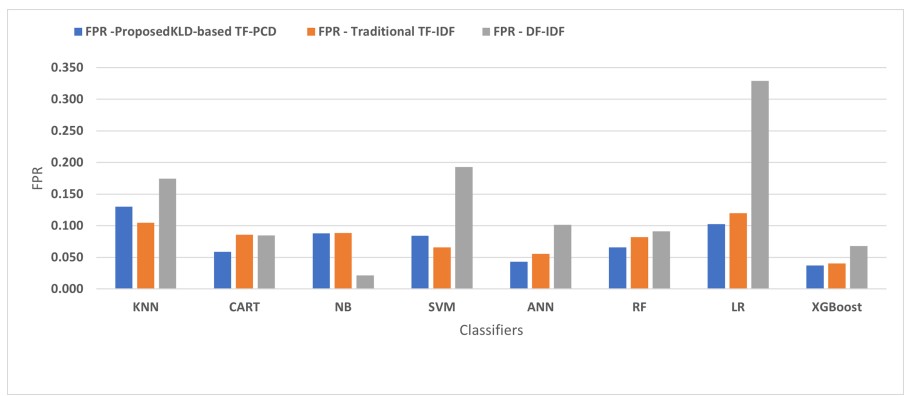

**Figure 5** The comparison of the False positive rate between the proposed KLD(TF-PDF)-based model and the related work-based models.

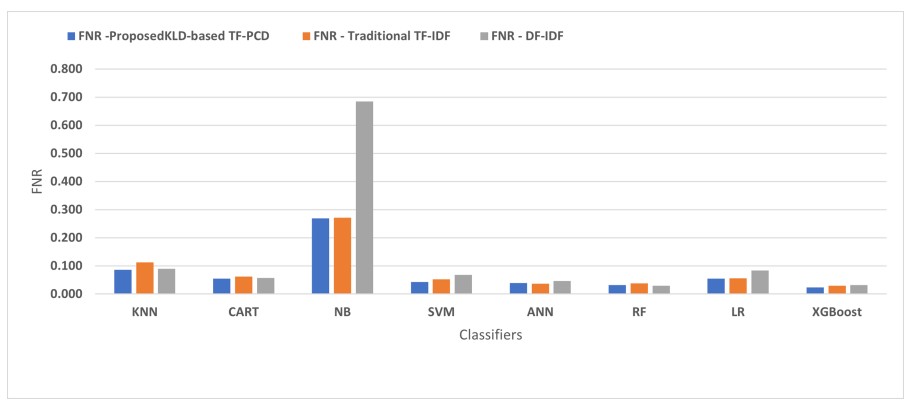

**Figure 6** The comparison of the False negative rate between the proposed KLD(TF-PDF)-based model and the related work-based models.

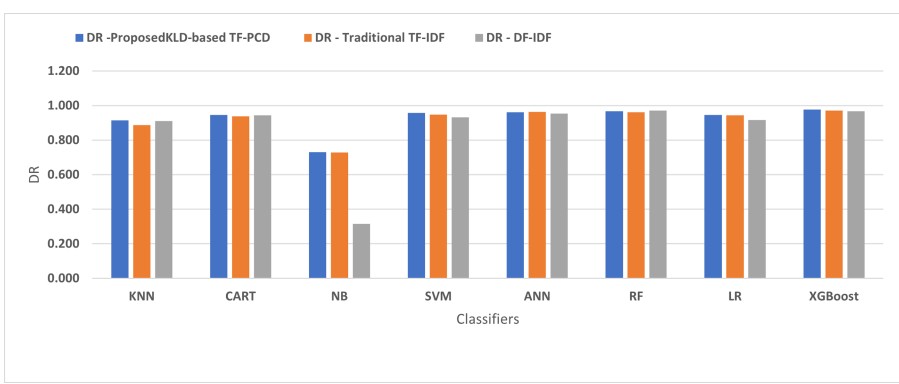

**Figure 7** The comparison of the Detection rate between the proposed KLD(TF-PDF)-based model and the related work-based models.

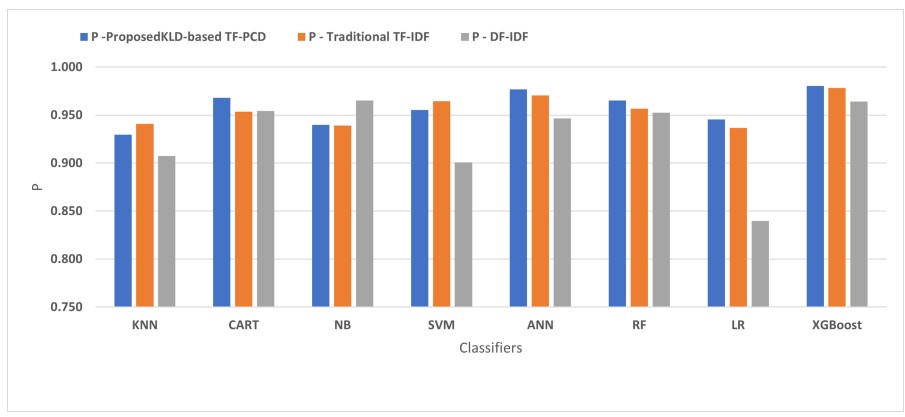

**Figure 8** The comparison of the Precision between the proposed KLD(TF-PDF)-based model and the related work-based models.

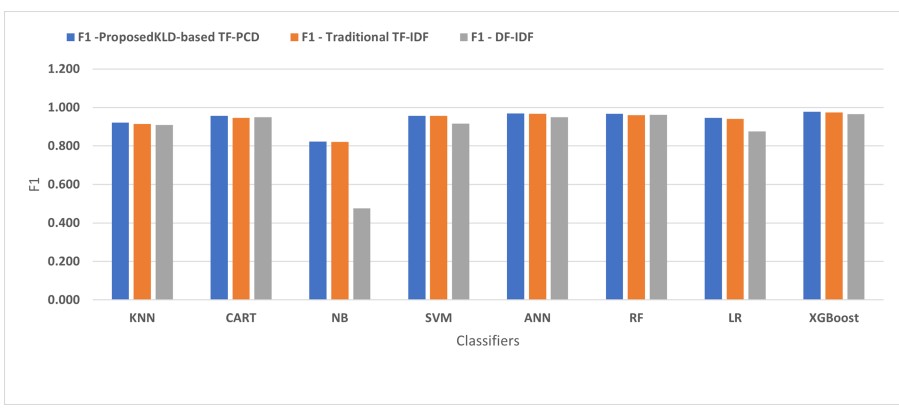

**Figure 9** The comparison of the *F*-measure between the proposed KLD(TF-PDF)-based model and the related work-based models.

## Significance test

To evaluate the significance of the proposed KLD-based TF-PCD algorithm statistically against the traditional TF-IDF and DF-IDF techniques based on the classification accuracy, the paired $t$-test has been carried out with the help 10-fold—cross-validation method using the XGBoost classifier. ($\alpha = 0.5$) is identified as the standard value to the degree of significance. In the paired $t$-test, two hypotheses are evaluated against each other. The first hypothesis supposes that the proposed algorithm and the techniques developed in the related work have the same detection accuracy, whereas the second hypothesis supposes that the proposed and related techniques have different detection accuracy. The first hypothesis is discarded if $(p\text{-value} < \alpha)$. Table 4 displays the $t$-test outcomes at the 95% level of significance. It can be observed in Table 4 that the classification accuracy of the KLD-based TF-PCD algorithm significantly enhanced the detection accuracy against the traditional TF-IDF and DF-IDF techniques based on the $p$-values obtained against both

**Table 4** *t*-test for KLD-based TF-PCD algorithm. Results of *t*-test for KLD-based TF-PCD algorithm against the traditional TF-IDF and DF-IDF techniques on classification accuracy.

| Representation technique | *t*-value | *p*-value | Significance |
|---|---|---|---|
| The proposed KLD-based TF-PCD/Traditional TF-IDF | 1.8595 | 0.0068 | Significant |
| The proposed KLD-based TF-PCD/DF-IDF developed by *Xue et al. (2019)* | 1.8595 | 0.0010 | Significant |

TF-IDF and DF-IDF techniques. The rate of accuracy enhancement is indicated in the *t*-values by which the significance is estimated. The results of the *t*-test show a significant improvement in the efficiency of the proposed KLD-based TF-PCD feature representation algorithm when compared to the traditional TF-IDF and DF-IDF techniques.

## ANALYSIS AND DISCUSSION

This study introduced the concept of representing malicious behaviors using weights which were influenced by how differently those behaviors were distributed in each class. The KLD-based TF-PCD algorithm was suggested and implemented to accurately calculate the weight-based vectors for the extracted features, including the sharing features, in order to reduce the model confusion level and thus improve the classification accuracy through decreasing the FPR and FNR. The proposed technique consists of two components, the Term Frequency TF component, and the Kullback-Liebler Divergence component. TF was employed in the proposed algorithm to compute the appearance ratio of each feature in the concerned document, while the KLD component was utilized to measure the difference between the distributions of each feature in malware and benign classes. Unlike previous work, which relies on feature distributions in entire documents without taking into account the feature class of those documents, the proposed representation algorithm measures the difference between the feature distributions in malware and benign classes to enrich the features that reflect big distribution difference values according to how big the difference value is, instead of assigning similar weights to those features based on their distribution in the all the documents.

Table 3 displays that the majority of the classifiers provided close classification performance except for KNN and NB classifiers, which experienced classification performance degradation. This reduction in the classification performance may be due to the nature of KNN and NB classifiers. Furthermore, the NB classifier assumes that each individual feature contributes independently of other features when the classification probability of testing data is calculated, while the KNN classifier classifies the testing data based on the K most similar instances (neighbors) in the training data, which are identified using Euclidean distance. Therefore, if there are several instances in the training data that provide the smallest distance from the testing data, the first instance is utilized to specify the class of testing data (*Gupta & Rani, 2020*).

Moreover, Figs. 4 and 9 illustrated that the performance of classification (accuracy and F1) for all the classifiers that were trained based on weights generated using the proposed KLD-based TF-PCD algorithm was higher than the same classifiers when they were trained utilizing weights calculated by the traditional TF-IDF and DF-IDF techniques, while Figs. 5,

6, 7 and 8 show that the classification performance in terms of FPR, FNR, DR, and P for the majority of classifiers with the proposed KLD-based TF-PCD algorithm was higher than the classification performance when the classifiers were developed using the traditional TF-IDF and DF-IDF techniques. This means that the proposed KLD-based TF-PCD algorithm was eligible to represent malicious behavior characteristics better than the other feature representation techniques. This is ascribed to the capability of the KLD-based TF-PCD algorithm to enhance the discrimination degree of the features that exist in almost all the documents and thus, in both malware and benign classes. However, those features can be more meaningful when the proposed KLD-based TF-PCD algorithm weighs them with the help of the differences in their probability distributions between benign and malware classes. Therefore, the proposed KLD-based TF-PCD algorithm benefits from the various distribution nature of the sharing features in malware and benign classes instead of considering those features as non-discrimination features with other feature representation techniques. As a result, new meaningful characteristics have been added by the proposed algorithm to promote the learned knowledge of the classifiers, and thus increase their ability to classify malicious behaviors accurately.

This implies that the mimic legitimate behaviors and the evasion techniques that were performed by malware and benign samples, respectively, had been represented using accurate weights. The intuition is that, while those behaviors appeared in most of the documents, malicious or benign, using the differences between their distributions in malware and benign classes made those behaviors closer to representing accurately whether malicious or legitimate activities since such behaviors can be found in both classes but in different frequency.

Although the proposed algorithm achieves a slight improvement rate in terms of accuracy, 0.51% compared to the TF-IDF techniques and 1.78% against the DF-IDF technique, the paired $t$-test prove that the proposed algorithm carried out a significant improvement. The improvement was achieved because the proposed algorithm generated new learning knowledge by which the developed model gained more ability to correctly classify the testing data. Additionally, as a result of the proposed algorithm carrying out more refinement for the weights that belong to the overlapping features, both FPR and FNR have been decreased.

## CONCLUSION

In this paper, the Kullback-Liebler Divergence-based Term Frequency-Probability Class Distribution (KLD-based TF-PCD) algorithm was proposed to represent the extracted features as weight-based vectors. KLD-based TF-PCD algorithm was developed to mitigate the limitations of the existing representation techniques where the sharing features were represented using inaccurate weights which were generated without considering the feature distributions in each class. KLD-based TF-PCD algorithm enriches the weight of each feature by measuring the difference between the probability distributions of that feature in malware and benign classes using the Kullback-Liebler Divergence tool. The feature's weight is higher when there is a significant difference between the probability

distributions of that feature in the malicious and benign classes. Therefore, the overlapping features that existed in both malware and benign classes but with different frequencies can be meaningful since the corresponding weights of these features were influenced by how they were differently distributed in malware and benign classes. Such an algorithm can benefit from the sharing features such as the ones that reflect the mimic legitimate behaviors performed by malware and the evasion behaviors achieved by benign. The proposed KLD-based TF-PCD algorithm was able to generate more accurate weights to represent the extracted features than the existing techniques. Eight classifiers, namely, k-nearest neighbor (KNN), regression trees (CART), Naive Bayes (NB), support vector machine (SVM), artificial neural network (ANN), random forest (RF), logistic regression (LR), and eXtreme Gradient Boosting (XGBoost) were implemented to evaluate the classification ability of the features which were represented using the proposed algorithm. A comparison between the proposed algorithm and the existing representation technique was also conducted. The comparison results show that the proposed feature representation algorithm represents the extracted features more accurately than the existing feature representation techniques. To assess the significance of the improvement for the proposed technique, a $t$-test was employed. The results of the $t$-test show that the improvement of the proposed algorithm against the traditional TF-IDF and DF-IDF techniques was significant, with a $t$-value of 1.8595 *versus* both techniques and $p$-values of 0.0068 and 0.0010 against the traditional TF-IDF and DF-IDF techniques, respectively. However, this study has limitations from which future work can be planned. The proposed KLD-based TF-PCD algorithm is established to be appropriate for binary classification tasks. Therefore, the complementary development by which the KLD-based TF-PCD algorithm can be adequate for the multiclassification task is interesting for our future work. In addition, the nature, and the position of the TF component in the KLD-based TF-PCD equation may be the root causes behind increasing the sparsity vector rate, which we plan to address in our future work. Although the experiments in this study show that applying SMOTE slightly decreases the classification accuracy, more investigation is needed to study the impact of the imbalanced dataset on the classification performance.

### Funding

This research work is funded by the Princess Nourah bint Abdulrahman University Researchers Supporting Project (PNURSP2023R40), Princess Nourah bint Abdulrahman University, Riyadh, Saudi Arabia. The funders had no role in study design, data collection and analysis, decision to publish, or preparation of the manuscript.

### Grant Disclosures

The following grant information was disclosed by the authors:
Princess Nourah bint Abdulrahman University Researchers: PNURSP2023R40.
Princess Nourah bint Abdulrahman University, Riyadh, Saudi Arabia.

## Competing Interests

Saeed Faisal is an Academic Editor for PeerJ.

## Author Contributions

- Faitouri A. Aboaoja conceived and designed the experiments, performed the experiments, analyzed the data, performed the computation work, prepared figures and/or tables, authored or reviewed drafts of the article, and approved the final draft.
- Anazida Zainal conceived and designed the experiments, performed the computation work, prepared figures and/or tables, and approved the final draft.
- Fuad A. Ghaleb conceived and designed the experiments, performed the computation work, prepared figures and/or tables, and approved the final draft.
- Norah Saleh Alghamdi performed the experiments, authored or reviewed drafts of the article, and approved the final draft.
- Faisal Saeed performed the experiments, authored or reviewed drafts of the article, and approved the final draft.
- Husayn Alhuwayji analyzed the data, authored or reviewed drafts of the article, and approved the final draft.

## Data Availability

The code and dataset are available in the Supplemental Files.

## Supplemental Information

Supplemental information for this article can be found online at http://dx.doi.org/10.7717/peerj-cs.1492#supplemental-information.

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
