# Peer review of "A Kullback-Liebler divergence-based representation algorithm for malware detection"

_PeerJ Computer Science, doi:10.7717/peerj-cs.1492_

## Round 0.1 · original submission · Major Revisions

The reviewers have substantial concerns about this manuscript. The authors should provide point-to-point responses to address all the concerns and provide a revised manuscript with the revised parts being marked in different color.

Reviewer 1 ·

Basic reporting

1. This manuscript has a good and comprehensive summary of the background information regarding the related literatures in this field.

2. The figures are blur and not able to read clearly.

3. For figure 1, it looks like some snapshot from other resource, this is prohibit in publication, please create your own figure.

4. For figures 4-9, the values of each column cannot be read.

5. All equations are messed up, cannot read equation content mostly, same with explanation of equations.

Experimental design

1. The dataset used in this study only contains 7208 malware ones and 3848 benign ones, this dataset itself is not enough.

2. And the dataset class distribution is biased, so more discussions should be made for how to solve this biased issue.

3. A 60% vs 40% for training and testing data is not suitable for such a small dataset. This should be change or add discussion to proof this is valid.

Validity of the findings

1. From the figures 4-9, seems like there's no difference between new introduced model and traditional model, so more discussions should be made here to explain how to proof the model introduced here is an improvement.

2. It is unclear which classifier data is used for significance test, and it is not clear how the values come out in the significance test, so cannot valid for the conclusion.

Additional comments

Please regenerate all figures and equations.

Reviewer 2 ·

Basic reporting

1. Clear and unambiguous, professional English used throughout.

I think there are plenty of paragraph/sentences can be improved. To name a few: "CNN convolutional neural network" the following sentence using a different convention "random forest (RF)". It will be better to keep the naming convention consistent. In addition, for line 601 to 602 "As long as the difference between the probability distributions in malware and benign classes is large, the weight of the feature is greater." and line 603 to line 607, the language could benefit from further refinement and clarity.

Use Capital letters in line 32, 28 the start of a sentence.

2. Professional article structure, figures, tables. Raw data shared.

There are a few equations that not shows correctly. Moreover, additional formatting of the figure would enhance its readability

Experimental design

The experiment design of the paper appears to be rigorous and performed to a high technical and ethical standard. The research question is well defined, relevant, and meaningful, and it is stated how the research fills an identified knowledge gap. The methods are described with sufficient detail and information to replicate.

Validity of the findings

The conclusion of the paper are well stated, linked to the original research question, and limited to supporting results. The paper compared eight classifiers, namely, K-Nearest Neighbor (KNN), Regression Trees (CART), Naive Bayes (NB), Support Vector Machine (SVM), Artificial Neural Network (ANN), Random Forest (RF), Logistic Regression (LR), and eXtreme Gradient Boosting (XGBoost) to evaluate the classification ability of the features which were represented using the proposed algorithm. The thorough analysis and discussion presented in the paper contribute to a convincing conclusion.

Additional comments

To summarize, the area of improvement of the paper (language improvement, equations format, figure format) outweighs the metrics in analysis section. I believe the paper has the potential to become excellent if the author resolved the comments raised.

Reviewer 3 ·

Basic reporting

1. There are many unknown symbols in the presented equations, for example, in equations 1, 2, and 6. It is impossible to understand the section of “Feature Representation”.
2. There are many grammar issues and misspelling issues in the current manuscript, the authors should go through the content carefully to fix them.
3. The table legends of Tables 2 and 3 are problematic. They were all presented as “Table 1”.

Experimental design

1. In the section of “Dataset”, the authors first introduced that the evasive malware samples and benign samples were from three studies, then the authors mentioned that “our experiments are carried out using a dataset that was used in our earlier work (Aboaoja et al., 2023)”. What exactly is the dataset that they used for experiments? Please give more explanations.

Validity of the findings

1. For the section of “Performance Measures”, the authors introduced precision as one of the validation measures. It will be good to further include recall as another validation measure to compare with precision. The trade-off between precision and recall could offer more information about the performance of the algorithm.
2. For the performance comparison results presented in figures 4, 5, 6, 7, 8, and 9, the performance improvement of KLD-based TF-PCD compared with traditional TF-IDF was too moderate, which makes the current study meaningless.

---

## Round 0.2 · accepted · Accept

The authors have addressed all concerns. I suggest accepting this manuscript.

Reviewer 1 ·

Basic reporting

no comment

Experimental design

no comment

Validity of the findings

no comment

Additional comments

The authors have responded well to the comments, the manuscript can be published as is.

Reviewer 2 ·

Basic reporting

N/A

Experimental design

N/A

Validity of the findings

N/A

Additional comments

The revision addressed all the feedbacks. No further comments.